# PKA regulation of neuronal function requires the dissociation of catalytic subunits from regulatory subunits

**Weihong Xiong, Maozhen Qin, Haining Zhong***

Vollum Institute, Oregon Health and Science University, Portland, United States

## eLife Assessment

This **important** paper demonstrates that different PKA subtypes exhibit distinct subcellular localization at rest in CA1 neurons. The authors provide **compelling** evidence that when all tested PKA subtypes are activated by norepinephrine, catalytic subunits translocate to dendritic spines but regulatory subunits remain unmoved. Furthermore, PKA-dependent regulation of synaptic plasticity and transmission can be supported only by wildtype, dissociable PKA, but not by inseparable PKA.

**\*For correspondence:**
zhong@ohsu.edu

**Competing interest:** The authors declare that no competing interests exist.

**Abstract** Protein kinase A (PKA) plays essential roles in diverse cellular functions. However, the spatiotemporal dynamics of endogenous PKA upon activation remain debated. The classical model predicts that PKA catalytic subunits dissociate from regulatory subunits in the presence of cAMP, whereas a second model proposes that catalytic subunits remain associated with regulatory subunits following physiological activation. Here, we report that different PKA subtypes, as defined by the regulatory subunit, exhibit distinct subcellular localization at rest in CA1 neurons of cultured hippocampal slices. Nevertheless, when all tested PKA subtypes are activated by norepinephrine, presumably via the β-adrenergic receptor, catalytic subunits translocate to dendritic spines but regulatory subunits remain unmoved. These differential spatial dynamics between the subunits indicate that at least a significant fraction of PKA dissociates. Furthermore, PKA-dependent regulation of synaptic plasticity and transmission can be supported only by wildtype, dissociable PKA, but not by inseparable PKA. These results indicate that endogenous PKA regulatory and catalytic subunits dissociate to achieve PKA function in neurons.

## Introduction

Cyclic adenosine monophosphate (cAMP)-dependent kinase, or protein kinase A (PKA), regulates diverse critical functions in nearly all mammalian cells, including neurons. PKA is a tetrameric protein consisting of two regulatory subunits (PKA-Rs) and two catalytic subunits (PKA-Cs; *Francis and Corbin, 1994*; *Johnson et al., 2001*). In the inactive state, each PKA-R binds to and inhibits a PKA-C. Binding of cAMP to PKA-R activates PKA-C. However, there are different proposals on the molecular events that follow activation.

For decades, PKA-C is thought to dissociate from PKA-R upon cAMP binding (*Beavo et al., 1974*; *Francis and Corbin, 1994*; *Gold, 2019*; *Johnson et al., 2001*; *Reimann et al., 1971*). Freed PKA-C molecules then move to phosphorylate their substrates. However, several studies (reviewed in *Gold, 2019*), including two notable recent publications (*Smith et al., 2017*; *Smith et al., 2013*), propose an alternative model, in which physiological concentrations of cAMP can activate PKA-C but do not result in its dissociation from PKA-R. Testing these two models will not only elucidate the biophysical

mechanism of PKA activation, but also have distinct implications in how PKA may achieve its specificity, which is thought to rely on spatial compartmentalization (*Wong and Scott, 2004*).

We have previously found that the majority of type IIβ PKA, as defined by PKA-R, is anchored to microtubules in the dendritic shaft of hippocampal CA1 pyramidal neurons where PKA-RIIβ is bound to the abundant microtubule associated protein MAP-2 (*Zhong et al., 2009*). Upon activation of the β-adrenergic receptor with norepinephrine, a fraction of PKA-C dissociated from PKA-RIIβ (*Tillo et al., 2017*). The freed PKA-C redistributed into dendritic spines, whereas PKA-RIIβ remained anchored at the dendritic shaft (*Tillo et al., 2017*; *Xiong et al., 2021*). These results are consistent with the classical PKA activation model. However, recent studies suggest that PKA-Rs other than PKA-RIIβ may be the more abundant isoforms in CA1 neurons (*Church et al., 2021*; *Ilouz et al., 2017*; *Weisenhaus et al., 2010*). It remains untested whether these PKA isoforms dissociate upon physiologically relevant stimulations in neurons.

Here, we examined whether PKA-C dissociates from all major PKA-R isoforms in CA1 neurons. The rescue of function following knockdown of PKA-C was compared between wildtype dissociable PKA

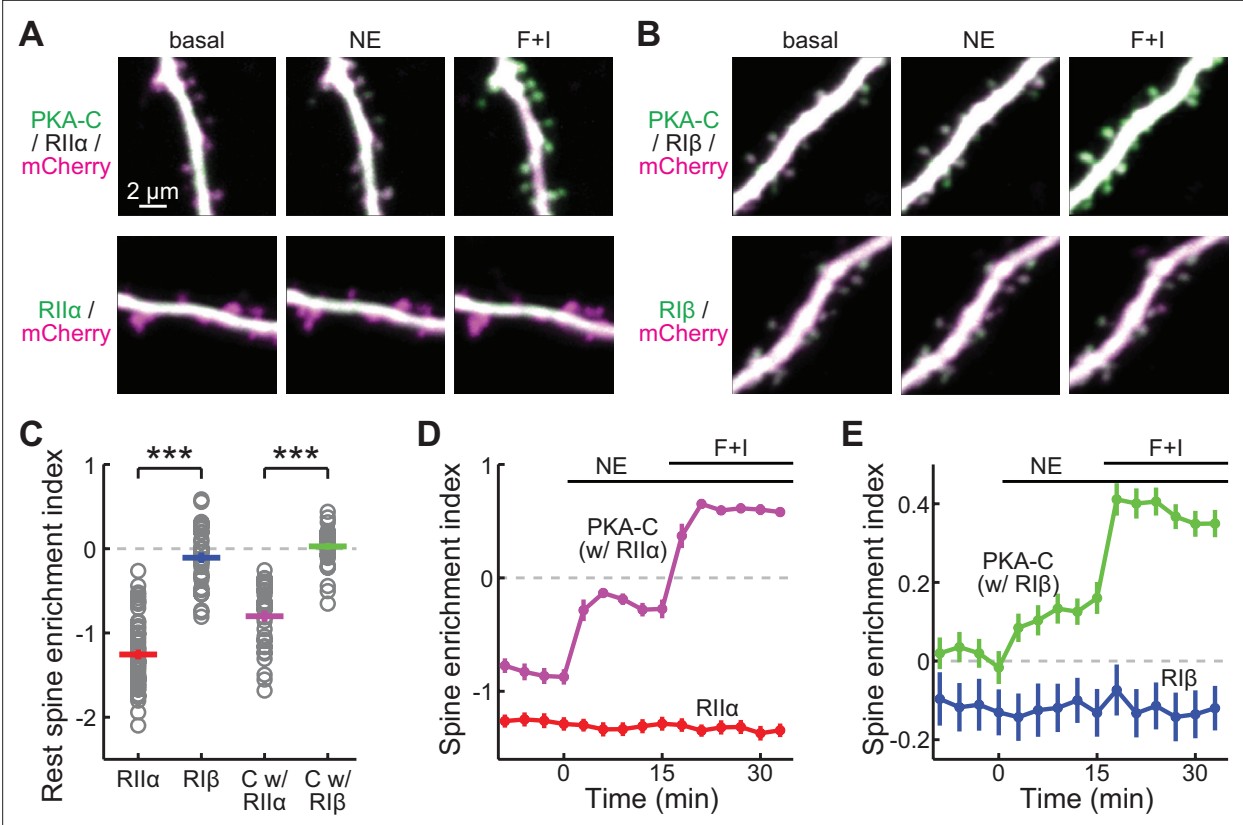

**Figure 1.** PKA-C but not PKA-R redistributes to spines upon activation. (**A, B**) Representative two-photon images of PKA-C-mEGFP co-expressed with PKA-RIIα or PKA-RIβ at rest, or in the presence of norepinephrine (NE) or forskolin and IBMX (*F+I*). mCherry (magenta) was co-expressed to reveal the neuronal morphology. (**C–E**) Quantification and comparison of the spine enrichment index at the resting state (**C**) and upon activation (**D, E**). As in panel C from left to right, n (spines/neurons)=53/11, 34/7, 33/6, and 36/7. Error bars represent s.e.m.

The online version of this article includes the following source data and figure supplement(s) for figure 1:

**Source data 1.** Numeric data for *Figure 1*.

**Figure supplement 1.** Spine enrichment indexes and the movement of PKA-C into spines upon activation are not dependent on the protein expression level.

**Figure supplement 1—source data 1.** Numeric data for *Figure 1—figure supplement 1*.

**Figure supplement 2.** PKA-C translocation can be driven by norepinephrine at low concentrations.

**Figure supplement 2—source data 1.** Numeric data for *Figure 1—figure supplement 2*.

**Figure supplement 3.** PKA-C and PKA-RIα differentially re-distributed upon activation.

**Figure supplement 3—source data 1.** Numeric data for *Figure 1—figure supplement 3*.

and an inseparable PKA variant in which PKA-C is covalently linked to PKA-R. The results support the classical model of PKA activation via dissociation.

## Results

Recent studies have suggested that PKA-RIIα and PKA-RIβ may be the prevalent isoforms in hippocampal CA1 neurons. Therefore, we co-expressed C-terminally monomeric EGFP tagged PKA-C (PKA-C-mEGFP; *Tillo et al., 2017*; *Zhong et al., 2009*) and a cytosolic marker (mCherry) with either PKA-RIIα or PKA-RIβ in CA1 neurons of organotypic hippocampal slice cultures (*Figure 1A and B*, upper left panels). At rest, PKA-C-mEGFP exhibited a distribution that was dependent on the co-expressed PKA-R: when co-expressed with PKA-RIIα, PKA-C was enriched in dendritic shafts; when co-expressed with PKA-RIβ, PKA-C was more evenly distributed (quantified using the spine enrichment indexes, or SEI, see Materials and methods; *Figure 1C*). This distribution was independent of the expression level (rest conditions in *Figure 1—figure supplement 1*) and largely resembled that of the corresponding PKA-R, as visualized using expressed PKA-R-mEGFP in separate experiments (*Figure 1A–C*).

Notably, upon application of norepinephrine (10 µM), PKA-C of both subtypes translocated to dendritic spines, but the subcellular localization of PKA-Rs remained unchanged (*Figure 1D and E*). A 5 x lowered norepinephrine concentration (2 µM) also resulted in similar dynamics of PKA-C (*Figure 1—figure supplement 2*), indicating that the PKA-C translocation happens in a wide range of neuromodulator concentrations. This differential re-distribution between PKA-C and PKA-R was more prominent following activation with forskolin (25 µM) and IBMX (50 µM). The translocation of PKA-C was independent of the expression level and the effect remained when extrapolating to the zero-overexpression level using a linear fit (*Figure 1—figure supplement 1*). These results can only be explained if at least a fraction of PKA-C dissociated from both PKA-RIIα and PKA-RIβ when a physiological stimulant was used. The differential re-distribution between PKA-C and PKA-R was also observed when PKA-RIα was used (*Figure 1—figure supplement 3*), although it was less implicated in hippocampal neurons. Together with our earlier results regarding PKA-C/PKA-RIIβ, we conclude that PKA-C dissociates from PKA-R with physiologically relevant stimuli.

Next, we asked whether PKA regulation of neuronal function is dependent on the dissociation of PKA-C from PKA-R. A key experiment supporting the non-dissociating PKA activation model was that PKA regulation of cell growth could be sustained by a construct in which PKA-C was fused to PKA-RIIα in one polypeptide chain via a flexible linker (named R-C; *Figure 2A*; *Smith et al., 2017*). We therefore asked whether this R-C construct could support PKA regulation of neuronal function. When

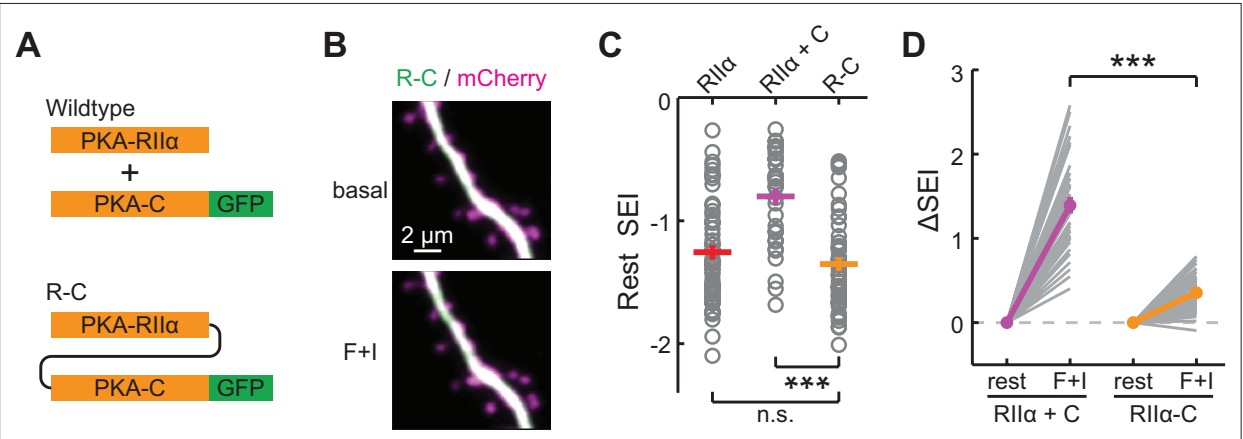

**Figure 2.** Characterization of the inseparable R-C. (**A**) Schematic of wildtype PKA versus R-C. In both cases PKA-C was C-terminally tagged by mEGFP. (**B–C**) Representative images (**B**), quantifications of resting distribution (**C**), and the distribution change upon stimulation by forskolin and IBMX (**D**) of R-C compared to PKA-RIIα-mEGFP and co-expressed PKA-C-mEGFP/PKA-RIIα. RIIα and RIIα+C data are from *Figure 1C*. n (spines/neurons)=48/10. Error bars represent s.e.m.

The online version of this article includes the following source data for figure 2:

**Source data 1.** Numeric data for *Figure 2*.

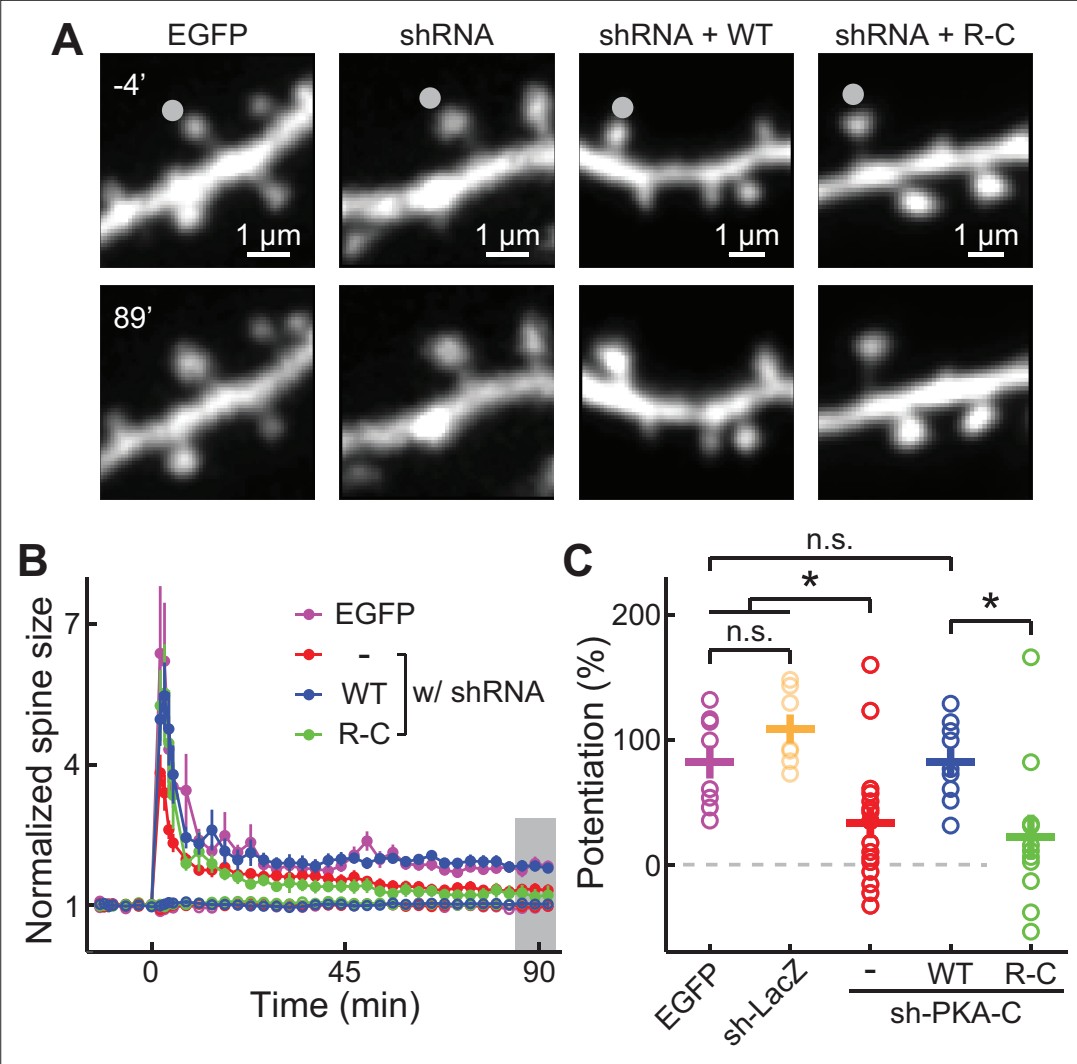

**Figure 3.** PKA regulation of synaptic plasticity cannot be sustained by an inseparable PKA variant. (A–C) Representative image (**A**), time course (**B**), and the degree of potentiation (**C**) at the indicated timepoints in panel B of single-spine LTP experiments as triggered by focal glutamate uncaging at the marked spines (gray dot). In panel B, both stimulated spines (solid circles) and non-stimulated control spines (open circles) are shown. As in panel C from left to right, n (spines, each from a different neuron)=8, 7, 17, 11, 9. Error bars represent s.e.m.

The online version of this article includes the following source data for figure 3:

**Source data 1.** Numeric data for *Figure 3*.

---

R-C-mEGFP was expressed in CA1 neurons, this construct exhibited a distribution highly similar to that of RIIα (*Figure 2B and C*). The tendency of this construct to translocate to the spine was largely diminished compared to PKA-C-mEGFP co-expressed with wildtype, unlinked PKA-RIIα (*Figure 2D*), indicating that the catalytic subunit in R-C was indeed inseparable from the regulatory subunit.

To evaluate the function of R-C, a previously established shRNA construct was used to selectively knock down PKA-Cα in CA1 neurons in cultured hippocampal slices (*Tillo et al., 2017*). Given that PKA activation is required for the late phase of long-term potentiation (L-LTP; *Abel et al., 1997*), we examined the structural LTP of individual dendritic spines of CA1 neurons elicited by focal two-photon glutamate uncaging (*Figure 3*; *Matsuzaki et al., 2004*). The shRNA knockdown of PKA-C resulted in attenuated LTP at 90 min after induction (*Figure 3A–C*). This attenuation was not observed when a control shRNA against LacZ was expressed (*Figure 3C*). The attenuated structural LTP was rescued by co-expression of shRNA-resistant wild-type PKA-C-mEGFP together with PKA-RIIα (*Figure 3A–C*).

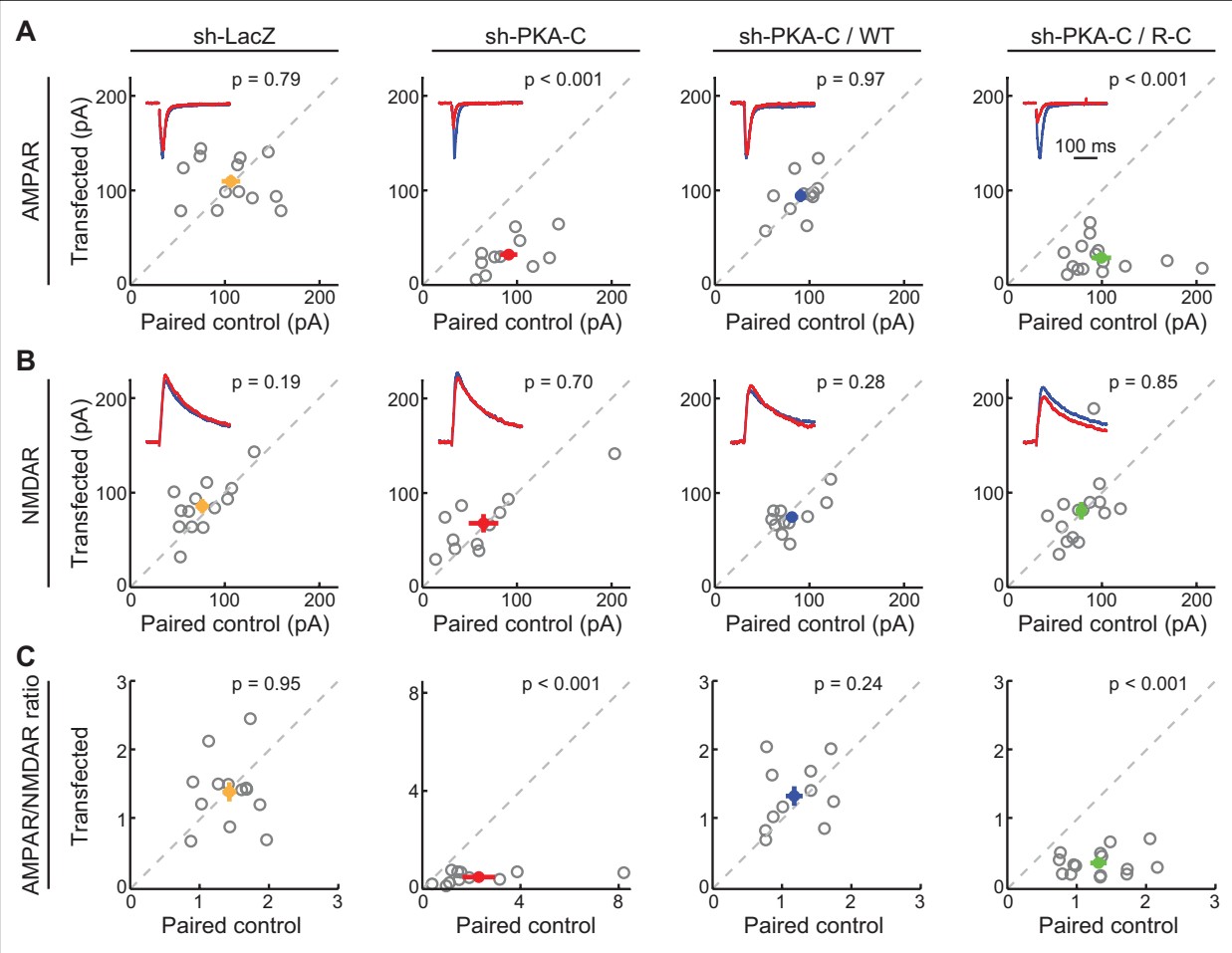

**Figure 4.** AMPA receptor-mediated synaptic transmission requires wildtype dissociable PKA. (**A–C**) Representative traces (red) normalized to the paired control (blue) (insets) and scatter plots of paired AMPA (**A**) and NMDA (**B**) receptor currents and AMPA/NMDA receptor current ratios (**C**) from neighboring untransfected CA1 neurons and those transfected with shRNA against PKA-C and the indicated shRNA-resistant rescue constructs. Statistical p values were obtained using a sign test (MATLAB). From left to right, n (neuron pairs)=13, 11, 11, and 15. Error bars represent s.e.m.

The online version of this article includes the following source data for figure 4:

**Source data 1.** Numeric data for *Figure 4*.

However, the R-C construct in which PKA-C was also resistant to shRNA knockdown but could not leave PKA-RIIα failed to rescue the phenotype.

PKA activity has also been shown to regulate synaptic transmission. We therefore examined evoked AMPA and NMDA receptor (AMPAR and NMDAR, respectively) currents in paired, transfected and adjacent untransfected CA1 neurons in cultured hippocampal slices. As shown previously (*Tillo et al., 2017*; *Xiong et al., 2021*), neurons expressing the shRNA construct against PKA-C exhibited significantly lower AMPAR currents (*Figure 4A*), but not NMDAR currents (*Figure 4B*). As a result, the AMPAR/NMDAR current ratio was also reduced (*Figure 4C*). The deficits were rescued by co-expression of shRNA-resistant, wild-type dissociable PKA-C-mEGFP and PKA-RIIα (*Figure 4*). However, the inseparable R-C construct failed to rescue the phenotype. Taken together, the R-C construct did not support normal PKA-dependent synaptic function.

## Discussion

Our results indicate that at least a fraction of PKA-C molecules dissociate from all tested PKA-R isoforms, including Iβ, IIα and previously tested IIβ, when activated by physiological stimuli. Given that PKA activity increases by two orders of magnitude when dissociated from PKA-R (*Miyamoto et al.,*

*1969*), even a small fraction of PKA-C dissociation will result in a marked increase of PKA kinase activity. These results corroborate the observations in intact non-neuronal cells that used FRET imaging and biochemical measurements, respectively (*Walker-Gray et al., 2017*; *Zaccolo et al., 2000*). Furthermore, PKA-C that is covalently linked to PKA-RIIα cannot functionally replace wildtype PKA for normal neuronal transmission or plasticity. Although this R-C construct has been shown to functionally replace endogenous PKA in terms of supporting the growth of a heterologous cell line (*Smith et al., 2017*), it cannot support all necessary PKA functions. Overall, we conclude that PKA-C dissociation from PKA-R is essential for PKA regulation of neuronal function. Additionally, PKA specificity is mediated by spatial compartmentalization (*Wong and Scott, 2004*). This is likely mediated by mechanism downstream of PKA dissociation, such as membrane tethering of freed PKA-C or its buffering by extra PKA-Rs (*Gaffarogullari et al., 2011*; *Tillo et al., 2017*; *Walker-Gray et al., 2017*; *Zhang et al., 2015*).

In addition, this study shows that PKA distribution in neuronal dendrites at the resting state is subtype dependent. PKA-RIIα is enriched in dendritic shaft in a way similar to PKA-RIIβ, likely via its interaction with the abundant microtubule binding protein MAP2 (*Tillo et al., 2017*; *Vallee et al., 1981*). Note that this observation does not exclude the importance a small fraction of PKA being anchored to synaptic sites via other PKA binding proteins. In contrast, PKA-RIα and PKA-RIβ are more evenly distributed between spine and dendrites. These observations establish a spatial organizational base for understanding subtype-specific PKA function in neurons.

This study also demonstrates that PKA is essential for long-term structural LTP of individual spines. PKA has been shown to facilitate the induction of LTP (i.e. metaplasticity) (*Thomas et al., 1996*; *Zhong et al., 2009*), and is required for the maintenance of L-LTP (*Abel et al., 1997*), as assayed using electrophysiological recording of postsynaptic currents. However, it has been suggested that the structural and synaptic current changes may not be causally linked (*Kopec et al., 2006*). Our results fill the gap to show that PKA is also required for maintaining the late phase of structural LTP.

## Materials and methods

**Key resources table**

| Reagent type (species) or resource | Designation | Source or reference | Identifiers | Additional information |
|---|---|---|---|---|
| Strain, strain background (Rattus norvegicus, Sprague Dawley) | Sprague Dawley rat | Charles River | Strain Code 001; RRID: RGD_734476 | |
| Recombinant DNA reagent | PKA-Cα-mEGFP (plasmid) | Addgene | # 45524; RRID: Addgene_45524 | |
| Recombinant DNA reagent | PKA-RIα-mEGFP (plasmid) | Addgene | # 45525; RRID: Addgene_45525 | |
| Recombinant DNA reagent | PKA-RIβ-mEGFP (plasmid) | Addgene | # 45526; RRID: Addgene_45526 | |
| Recombinant DNA reagent | PKA-RIIα-mEGFP (plasmid) | Addgene | # 45527; RRID: Addgene_45527 | |
| Recombinant DNA reagent | PKA-RIα (plasmid) | This paper | | *Figure 1—figure supplement 3* |
| Recombinant DNA reagent | PKA-RIβ (plasmid) | This paper | | *Figure 1* |
| Recombinant DNA reagent | PKA-RIIα (plasmid) | Addgene | #168492; RRID: Addgene_168492 | |
| Recombinant DNA reagent | PKA-RIIα-PKA-Cα-mEGFP (plasmid) | This paper | | *Figure 2* |
| Recombinant DNA reagent | shPKA against PKA-Cα with DsRed co-expression (plasmid) | This paper | | The shRNA was developed in *Tillo et al., 2017*; *Figure 3* |
| Recombinant DNA reagent | shPKA against LacZ with DsRed co-expression (plasmid) | This paper | | *Figure 3* |
| Recombinant DNA reagent | mCherry2 (plasmid) | Addgene | #54517; RRID: Addgene_54517 | |

*Continued on next page*

*Continued*

| Reagent type (species) or resource | Designation | Source or reference | Identifiers | Additional information |
|---|---|---|---|---|
| Chemical compound, drug | Norepinephrine | Tocris | 5169 | |
| Chemical compound, drug | Forskolin | LC Labs | F-9926 | |
| Chemical compound, drug | IBMX | Sigma-Aldrich | I7018 | |
| Chemical compound, drug | MNI-glutamate | Tocris | 1490 | |
| Chemical compound, drug | TTX | Tocris | 1069 | |
| Chemical compound, drug | 2-Chloroadenosine | Sigma-Aldrich | C5134 | |
| Chemical compound, drug | GABAzine (SR 95531) | Tocris | 1262 | |
| Software, algorithm | MATLAB | MathWorks | RRID: SCR_001622 | |
| Software, algorithm | SI_View | *Zhong, 2022* | https://github.com/HZhongLab/SI_View | |

## Materials availability statement

All previously unpublished constructs and their sequences will be submitted to Addgene. All software is publicly accessible as indicated in the Key Resource Table.

## Plasmid constructs

Constructs were made using standard mutagenesis and subcloning methods. In the R-C construct, mouse PKA-RIIα and PKA-Cα were fused via a linker with residues WDPGSGSLEAGCKNFFPRSFTSCG SLEGGSAAA that were previously used (*Smith et al., 2017*).

## Organotypic hippocampal slice cultures and transfections

Cultured rat hippocampal slices were prepared from P6 – P8 (typically P7) pups, as described previously (*Stoppini et al., 1991*; *Zhong et al., 2009*). Animal experiments were performed in accordance with the Guide for the Care and Use of Laboratory Animals of the National Institutes of Health, and were approved by the Institutional Animal Care and Use Committee (IACUC) of the Oregon Health & Science University (#IP00002274). cDNA constructs were transfected after 1.5–3 weeks in vitro via the biolistic gene transfer method using the Helios gene gun and 1.6 μm gold beads (*Figure 1*) or, where long-term expression (~1 week) was required, with single-cell electroporation (*Figure 2*; *Otmakhov and Lisman, 2012*).

## Two-photon imaging and two-photon glutamate uncaging

A custom built two-photon microscope based on an Olympus BW51WI microscope body was used. Laser beams from two different Ti:Sapphire lasers (Maitai, Newport) were aligned to allow for simultaneous two-photon excitation and photoactivation. Laser intensities were controlled by Pockels cells (Conoptics). Imaging and photoactivation were controlled by ScanImage (Vidrio Tech) (*Pologruto et al., 2003*). Slices were perfused during imaging in gassed artificial cerebral spinal fluid (ACSF) containing (mM) 127 NaCl, 25 NaHCO$_3$, 25 D-glucose, 2.5 KCl, 4 MgCl$_2$, 4 CaCl$_2$, and 1.25 NaH$_2$PO$_4$ with 0.5 μM tetrodotoxin (TTX). mEGFP fluorescence (green) was unmixed from that of the cytosolic marker (mCherry or DsRed Express) using a dichroic (Chroma 565DCXR) and band-pass filters (Chroma HQ510/70 for green and Semrock FF01-630/92 for red).

For single-spine structural LTP experiments, 2.25 mM MNI-caged-L-glutamate (Tocris) was added to ACSF containing 4 mM calcium, 0.05 mM magnesium, 1 μM TTX and 4 μM 2-chloroadenosine, as

previously described (*Harvey et al., 2008*). To trigger structural plasticity, 30 pulses of 4 ms 16 mW (at back focal plane) 720 nm laser light were delivered to the spine head at 0.5 Hz.

Image analysis was performed using custom software written in MATLAB called *SI_View* (https://github.com/HZhongLab/SI_View; *Zhong, 2022*; *Ma et al., 2022*). Using the software, regions of interest (ROIs) were manually drawn to isolate spines or their immediately adjacent dendritic shaft. Only the spines well isolated from the dendrite laterally throughout the entire experiments were included. Spine enrichment index was calculated as:

$$SEI = \log_2[(F_{green}/F_{red})_{spine}/(F_{green}/F_{red})_{shaft}]$$

in which F is the average fluorescence intensity in an ROI.

The expression level was estimated by the maximal intensity from thick apical dendrite near the soma (typically 50–100 µm) similar to previously described (*Harvey et al., 2008*; *Tillo et al., 2017*). To minimize the influence of noise, a line of 9 pixels (pixel size ~0.04 µm) thick was manually drawn transecting the dendrite through the visually identified, brightest region. The line profile was further smoothened by 5 pixels along the line and baseline subtracted before determining the maximal value. To combine measurements from different hardware configuration (e.g. different microscopes), the data from each configuration and experiment were internally corrected for laser stimulation intensity and then normalized to the average of all data under the same condition.

## Electrophysiology

Whole-cell voltage-clamp recordings were performed using a MultiClamp 700B amplifier (Molecular Devices). Electrophysiological signals were filtered at 2 kHz and digitized and acquired at 20 kHz using custom software written in MATLAB. Slices were perfused with artificial cerebrospinal fluid containing 4 mM Ca and 4 mM Mg. The internal solution contained (in mM) 132 Cs-gluconate, 10 HEPES, 10 Na-phosphocreatine, 4 MgCl2, 4 Na2-ATP, 0.4 Na-GTP, 3 Na-ascorbate, 3 QX314, and 0.2 EGTA with an osmolarity of 295 mOsmol/kg. The junction potential was calculated to be –17 mV using a built-in function in the Clampfit software (Molecular Devices). Several less abundant anions (phosphocreatine, ATP, GTP and ascorbate) were omitted in the calculation due to lack of data in the program. The Cl reversal potential was –75 mV.

To reduce recurrent activities, cultured hippocampal slices were cut on both sides of CA1 and 4 µM 2-chloroadenosine (Sigma) was present in all recording experiments. 10 µM GABAzine (SR 95531, Tocris) was also included to suppress GABA currents. For electrical stimulation, a bipolar, θ-glass stimulating electrode (Warner Instruments) was positioned in the stratum radiatum 100–150 µm lateral to the recorded neuron. For all recordings, a transfected neuron and an untransfected neuron located within 50 µm of each other were sequentially recorded without repositioning the stimulation electrode. Measurements were carried out on averaged traces from approximately 20 trials under each condition. For AMPAR currents, the cells were held at –60 mV (before correcting for the junction potential) and the current was measured as the baseline-subtracted peak current within a window of 2–50ms after electric stimulation. For NMDAR currents, the average currents at 140–160ms after stimulation were used when the cells were held at +55 mV (before correcting for the junction potential).

## Data analysis, presentation, and statistics

Quantification and statistical tests were performed using custom software written in MATLAB. All experiments were replicated in multiple neurons across ≥3 slices from ≥2 transfections. No data were excluded unless the cell was lost during an experiment (the cell became blebby during imaging, or the seal was lost during recording). Averaged data are presented as mean ± s.e.m., unless noted otherwise. p values were obtained from one-way ANOVA tests, unless noted otherwise. In all figures, *: $p \leq 0.05$ and is considered statistically significant after Bonferroni correction for multiple tests, **: $p \leq 0.01$, and ***: $p \leq 0.001$.

## Acknowledgements

We thank all members of the Mao and Zhong laboratories at the Vollum Institute for constructive discussions. We thank Drs John Williams and Michael Muniak for critical comments and edits on

the manuscript. This work was supported by three NIH BRAIN Initiative awards (U01NS094247, RF1NS133599, and RF1MH130784) and an NINDS R01 grant (R01NS127013) to HZ.

## Additional information

### Funding

| Funder | Grant reference number | Author |
|---|---|---|
| National Institute of Neurological Disorders and Stroke | U01NS094247 | Haining Zhong |
| National Institute of Neurological Disorders and Stroke | RF1NS133599 | Haining Zhong |
| National Institute of Mental Health | RF1MH130784 | Haining Zhong |
| National Institute of Neurological Disorders and Stroke | R01NS127013 | Haining Zhong |

The funders had no role in study design, data collection and interpretation, or the decision to submit the work for publication.

### Author contributions

Weihong Xiong, Data curation, Formal analysis; Maozhen Qin, Resources; Haining Zhong, Conceptualization, Data curation, Formal analysis, Supervision, Funding acquisition, Writing - original draft, Writing - review and editing

### Author ORCIDs

Haining Zhong ⬤ https://orcid.org/0000-0002-7109-4724

### Ethics

Animal experiments were performed in accordance with the Guide for the Care and Use of Laboratory Animals of the National Institutes of Health, and were approved by the Institutional Animal Care and Use Committee (IACUC) of the Oregon Health & Science University (#IP00002274).

Reviewer #1 (Public review): https://doi.org/10.7554/eLife.93766.3.sa1
Reviewer #2 (Public review): https://doi.org/10.7554/eLife.93766.3.sa2
Reviewer #3 (Public review): https://doi.org/10.7554/eLife.93766.3.sa3
Author response https://doi.org/10.7554/eLife.93766.3.sa4

## Additional files

### Supplementary files

• MDAR checklist

### Data availability

All data generated or analyzed in this study are included in the manuscript and supporting files. Source data files are provided.

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
