## [Editor Report · eLife Assessment]

This **important** paper demonstrates that different PKA subtypes exhibit distinct subcellular localization at rest in CA1 neurons. The authors provide **compelling** evidence that when all tested PKA subtypes are activated by norepinephrine, catalytic subunits translocate to dendritic spines but regulatory subunits remain unmoved. Furthermore, PKA-dependent regulation of synaptic plasticity and transmission can be supported only by wildtype, dissociable PKA, but not by inseparable PKA.

---

## [Referee Report · Reviewer #1 (Public review)]

Summary:

This is a short self-contained study with a straightforward and interesting message. The paper focuses on settling whether PKA activation requires dissociation of the catalytic and regulatory subunits. This debate has been ongoing for ~ 30 years, with renewed interest in the question following a publication in Science, 2017 (Smith et al.). Here, Xiong et al demonstrate that fusing the R and C subunits together (in the same way as Smith et al) prevents the proper function of PKA in neurons. This provides further support for the dissociative activation model - it is imperative that researchers have clarity on this topic since it is so fundamental to building accurate models of localised cAMP signalling in all cell types. Furthermore, their experiments highlight that C subunit dissociation into spines is essential for structural LTP, which is an interesting finding in itself. They also show that preventing C subunit dissociation reduces basal AMPA receptor currents to the same extent as knocking down the C subunit. Overall, the paper will interest both cAMP researchers and scientists interested in fundamental mechanisms of synaptic regulation.

Strengths:

The experiments are technically challenging and well executed. Good use of control conditions e.g untransfected controls in Figure 4.

Weaknesses:

The novelty is lessened given the same team has shown dissociation of the C subunit into dendritic spines from RIIbeta subunits localised to dendritic shafts before (Tillo et al., 2017). Nevertheless, the experiments with RII-C fusion proteins are novel and an important addition.

---

## [Referee Report · Reviewer #2 (Public review)]

Summary:

PKA is a major signaling protein which has been long studied and is vital for synaptic plasticity. Here, the authors examine the mechanism of PKA activity and specifically focus on addressing the question of PKA dissociation as a major mode of its activation in dendritic spines. This would potentially allow to determine the precise mechanisms of PKA activation and address how it maintains spatial and temporal signaling specificity.

Strengths:

The results convincingly show that PKA activity is governed by the subcellular localization in dendrites and spines and is mediated via subunit dissociation. The authors make use of organotypic hippocampal slice cultures, where they use pharmacology, glutamate uncaging, and electrophysiological recordings.

Overall, the experiments and data presented are well executed. The experiments all show that at least in the case of synaptic activity, distribution of PKA-C to dendritic spines is necessary and sufficient for PKA mediated functional and structural plasticity.

The authors were able to persuasively support their claim that PKA subunit dissociation is necessary for its function and localization in dendritic spines. This conclusion is important to better understand the mechanisms of PKA activity and its role in synaptic plasticity.

Weaknesses:

While the experiments are indeed convincing and well executed, the data presented is similar to previously published work from the Zhong lab (Tillo et al., 2017, Zhong et al 2009). This reduces the novelty of the findings in terms of re-distribution of PKA subunits, which was already established, at least to some degree.

---

## [Referee Report · Reviewer #3 (Public review)]

Summary:

Xiong et al. investigated the debated mechanism of PKA activation using hippocampal CA1 neurons under pharmacological and synaptic stimulations. Examining all major PKA-R isoforms in these neurons, they found that a portion of PKA-C dissociates from PKA-R and translocate into dendritic spines following norepinephrine bath application. Additionally, their use of a non-dissociable form of PKA demonstrates its essential role in structural long-term potentiation (LTP) induced by two-photon glutamate uncaging, as well as in maintaining normal synaptic transmission, as verified by electrophysiology. This study presents a valuable finding on the activation-dependent re-distribution of PKA catalytic subunits in CA1 neurons, a process vital for synaptic functionality. The robust evidence provided by the authors makes this work particularly relevant for biologists seeking to understand PKA activation mechanisms, its downstream effects, and synaptic plasticity.

Strengths:

The study is methodologically robust, particularly in the application of two-photon imaging and electrophysiology. The experiments are well-designed with effective controls and a comprehensive analysis. The credibility of the data is further enhanced by the research team's previous works in related experiments. The study provides sufficient evidence to support the classical model of PKA activation via dissociation in neurons.

Weaknesses:

No specific weaknesses are noted in the current study; future research could provide additional insights by exploring PKA dissociation under varied physiological conditions, particularly in vivo, to further validate and expand upon these findings.

---

## [Author Response]

The following is the authors’ response to the original reviews.

New Experiments

(1) Activation-dependent dynamics of PKA with the RIα regulatory subunit, adding to the answer to Reviewers 1 and 2. To determine the dynamics of all PKA isoforms, we have added experiments that used PKA-RIα as the regulatory subunit. We found differential translocation between PKA-C (co-expressed with PKA-RIα) and PKA-RIα (Figure 1–figure supplement 3), similar to the results when PKA-RIIα or PKA-RIβ was used.

(2) PKA-C dynamics elicited by a low concentration of norepinephrine, addressing Reviewer 3’s comment. We have found that PKA-C (co-expressed with RIIα) exhibited similar translocation into dendritic spines in the presence of a 5x lowered concentration (2 μM) of norepinephrine, suggesting that the translocation occurs over a wide range of stimulus strengths (Figure 1-figure supplement 2).

**Reviewer #1 (Public Review):**
Summary:This is a short self-contained study with a straightforward and interesting message. The paper focuses on settling whether PKA activation requires dissociation of the catalytic and regulatory subunits. This debate has been ongoing for ~ 30 years, with renewed interest in the question following a publication in Science, 2017 (Smith et al.). Here, Xiong et al demonstrate that fusing the R and C subunits together (in the same way as Smith et al) prevents the proper function of PKA in neurons. This provides further support for the dissociative activation model - it is imperative that researchers have clarity on this topic since it is so fundamental to building accurate models of localised cAMP signalling in all cell types. Furthermore, their experiments highlight that C subunit dissociation into spines is essential for structural LTP, which is an interesting finding in itself. They also show that preventing C subunit dissociation reduces basal AMPA receptor currents to the same extent as knocking down the C subunit. Overall, the paper will interest both cAMP researchers and scientists interested in fundamental mechanisms of synaptic regulation.Strengths:The experiments are technically challenging and well executed. Good use of control conditions e.g untransfected controls in Figure 4.

We thank the reviewer for their accurate summarization of the position of the study in the field and for the positive evaluation of our study.

Weaknesses:The novelty is lessened given the same team has shown dissociation of the C subunit into dendritic spines from RIIbeta subunits localised to dendritic shafts before (Tillo et al., 2017). Nevertheless, the experiments with RII-C fusion proteins are novel and an important addition.

We thank the reviewer for noticing our earlier work. The first part of the current work is indeed an extension of previous work, as we have articulated in the manuscript. However, this extension is important because recent studies suggested that the majority of PKA-RIIβ are axonal localized. The primary PKA subtypes in the soma and dendrite are likely PKA-RIβ or PKA-RIIα. Although it is conceivable that the results from PKA-RIIβ can be extended to the other subunits, given the current debate in the field regarding PKA dissociation (or not), it remains important to conclusively demonstrate that these other regulatory subunit types also support PKA dissociation within intact cells in response to a physiological stimulant. To complete the survey for all PKA-R isoforms, we have now added data for PKA-RIα (New Experiment #1), as they are also expressed in the brain (e.g., https://www.ncbi.nlm.nih.gov/gene/5573). Additionally, as the reviewer points out, our second part is a novel addition to the literature.

**Reviewer #2 (Public Review):**
Summary:PKA is a major signaling protein that has been long studied and is vital for synaptic plasticity. Here, the authors examine the mechanism of PKA activity and specifically focus on addressing the question of PKA dissociation as a major mode of its activation in dendritic spines. This would potentially allow us to determine the precise mechanisms of PKA activation and address how it maintains spatial and temporal signaling specificity.Strengths:The results convincingly show that PKA activity is governed by the subcellular localization in dendrites and spines and is mediated via subunit dissociation. The authors make use of organotypic hippocampal slice cultures, where they use pharmacology, glutamate uncaging, and electrophysiological recordings.Overall, the experiments and data presented are well executed. The experiments all show that at least in the case of synaptic activity, the distribution of PKA-C to dendritic spines is necessary and sufficient for PKA-mediated functional and structural plasticity.The authors were able to persuasively support their claim that PKA subunit dissociation is necessary for its function and localization in dendritic spines. This conclusion is important to better understand the mechanisms of PKA activity and its role in synaptic plasticity.

We thank the reviewer for their positive evaluation of our study.

Weaknesses:While the experiments are indeed convincing and well executed, the data presented is similar to previously published work from the Zhong lab (Tillo et al., 2017, Zhong et al 2009). This reduces the novelty of the findings in terms of re-distribution of PKA subunits, which was already established. A few alternative approaches for addressing this question: targeting localization of endogenous PKA, addressing its synaptic distribution, or even impairing within intact neuronal circuits, would highly strengthen their findings. This would allow us to further substantiate the synaptic localization and re-distribution mechanism of PKA as a critical regulator of synaptic structure, function, and plasticity.

We thank the reviewer for noticing our earlier work. The first part of the current work is indeed an extension of previous work, as we have articulated in the manuscript. However, this extension is important because recent studies suggested that the majority of PKA-RIIβ are axonal localized. The primary PKA subtypes in the soma and dendrite are likely PKA-RIβ or PKA-RIIα. Although it is conceivable that the results from PKA-RIIβ can be extended to the other subunits, given the current debate in the field regarding PKA dissociation (or not), it remains important to conclusively demonstrate that these other regulatory subunit types also support PKA dissociation within intact cells in response to a physiological stimulant. To complete the survey for all PKA-R isoforms, we have now added data for PKA-RIα (New Experiment #1), as they are also expressed in the brain (e.g., https://www.ncbi.nlm.nih.gov/gene/5573). Additionally, as Reviewer 1 points out, our second part is a novel addition to the literature.

We also thank the reviewer for suggesting the experiments to examine PKA’s synaptic localization and dynamics as a key mechanism underlying synaptic structure and function. We agree that this is a very interesting topic. At the same time, we feel that this mechanistic direction is open ended at this time and beyond what we try to conclude within this manuscript: prevention of PKA dissociation in neurons affects synaptic function. Therefore, we will save the suggested direction for future studies. We hope the reviewer understand.

**Reviewer #3 (Public Review):**
Summary:Xiong et al. investigated the debated mechanism of PKA activation using hippocampal CA1 neurons under pharmacological and synaptic stimulations. Examining the two PKA major isoforms in these neurons, they found that a portion of PKA-C dissociates from PKA-R and translocates into dendritic spines following norepinephrine bath application. Additionally, their use of a non-dissociable form of PKC demonstrates its essential role in structural long-term potentiation (LTP) induced by two-photon glutamate uncaging, as well as in maintaining normal synaptic transmission, as verified by electrophysiology. This study presents a valuable finding on the activation-dependent re-distribution of PKA catalytic subunits in CA1 neurons, a process vital for synaptic functionality. The robust evidence provided by the authors makes this work particularly relevant for biologists seeking to understand PKA activation and its downstream effects essential for synaptic plasticity.Strengths:The study is methodologically robust, particularly in the application of two-photon imaging and electrophysiology. The experiments are well-designed with effective controls and a comprehensive analysis. The credibility of the data is further enhanced by the research team's previous works in related experiments. The conclusions of this paper are mostly well supported by data. The research fills a significant gap in our understanding of PKA activation mechanisms in synaptic functioning, presenting valuable insights backed by empirical evidence.

We thank the reviewer for their positive evaluation of our study.

Weaknesses:The physiological relevance of the findings regarding PKA dissociation is somewhat weakened by the use of norepinephrine (10 µM) in bath applications, which might not accurately reflect physiological conditions. Furthermore, the study does not address the impact of glutamate uncaging, a well-characterized physiologically relevant stimulation, on the redistribution of PKA catalytic subunits, leaving some questions unanswered.

We agreed with the Reviewer that testing under physiological conditions is critical especially given the current debate in the literature. That is why we tested PKA dynamics induced by the physiological stimulant, norepinephrine. It has been suggested that, near the release site, local norepinephrine concentrations can be as high as tens of micromolar (*Courtney and Ford, 2014*). Based on this study, we have chosen a mid-range concentration (10 μM). At the same time, in light of the Reviewer’s suggestion, we have now also tested PKA-RIIα dissociation at a 5x lower concentration of norepinephrine (2 μM; New Experiment #2). The activation and translocation of PKA-C is also readily detectible under this condition to a degree comparable to when 10 μM norepinephrine was used.

Regarding the suggested glutamate uncaging experiment, it is extremely challenging because of finite signal-to-noise ratios in our experiments. From our past studies, we know that activated PKA-C can diffuse three dimensionally, with a fraction as membrane-associated proteins and the other as cytosolic proteins. Although we have evidence that its membrane affinity allows it to become enriched in dendritic spines, it is not known (and is unlikely) that activated PKA-C is selectively targeted to a particular spine. Glutamate uncaging of a single spine presumably would locally activate a small number of PKA-C. It will be very difficult to trace the 3D diffusion of these small number of molecules in the presence of surrounding resting-state PKA-C molecules. Finally, we hope the reviewer agrees that, regardless of the result of the glutamate uncaging experiment, the above new experiment (New Experiment #2) already indicate that certain physiologically relevant stimuli can drive PKA-C dissociation from PKA-R and translocation to spines, supporting our conclusion.

**Reviewer #2 (Recommendations For The Authors):**
It was a pleasure reading your paper, and the results are well-executed and well-presented.My main and only recommendations are two ways to further expand the scope of the findings.First, I believe addressing the endogenous localization of PKA-C subunit before and after PKA activation would be highly important to validate these claims. Overexpression of tagged proteins often shows vastly different subcellular distribution than their endogenous counterparts. Recent technological advances with CRISPR/Cas9 gene editing (Suzuki et al Nature 2016 and Gao et al Neuron 2019 for example) which the Zhong lab recently contributed to (Zhong et al 2021 eLife) allow us to tag endogenous proteins and image them in fixed or live neurons. Any experiments targeting endogenous PKA subunits that support dissociation and synaptic localization following activation would be very informative and greatly increase the novelty and impact of their findings.

We agreed that addressing the endogenous PKA dynamics is important. However, despite recent progress, endogenous labeling using CRISPR-based methods remains challenging and requires extensive optimization. This is especially true for signaling proteins whose endogenous abundance is often low. We have tried to label PKA catalytic subunits and regulatory subunits using both the homologous recombination-based method SLENDR and our own non-homologous end joining-based method CRISPIE. We did not succeed, in part because it is very difficult to see any signal under wide-field fluorescence conditions, which makes it difficult to screen different constructs for optimizing parameters. It is also possible that, at the endogenous abundance, the label is just not bright enough to be seen. Nevertheless, for both PKA type Iβ and type IIα that we studied in this manuscript, we have correlated the measured parameters (specifically, Spine Enrichment Index or SEI) with the overexpression level (Figure 1-figure supplement 1). We found that they are not strongly correlated with the expression level under our conditions. By extrapolating to non-overexpression conditions, our conclusion remains valid.

To overcome the inability to label endogenous PKA subunits using CRISPR-based methods, we have also attempted a conditional knock-in method call ENABLED that we previously developed to label PKA-Cα. In preliminary results, we found that endogenously label PKA were very dim. However, in a subset of cells that are bright enough to be quantified, the PKA catalytic subunit indeed translocated to dendritic spines upon stimulation (see Additional Fig. 1 in the next page), corroborating our results using overexpression. These results, however, are not ready to be published because characterization of the mouse line takes time and, at this moment, the signal-to-noise ratio remains low. We hope that the reviewer can understand.

**Author response image 1. sa4fig1:** Endogeneous PKA-Cα translocate to dendritic spines upon activation.

Second, experiments which would advance and validate these findings in vivo would be highly valuable. This could be achieved in a number of ways - one would be overexpression of tagged PKA versions and examining sub-cellular distribution before and after physiological activation in vivo. Another possibility is in vivo perturbation - one would speculate that disruption or tethering of PKA subunits to the dendrite would lead to cell-specific functional and structural impairments. This could be achieved in a similar manner to the in vitro experiments, with a PKA KO and replacement strategy of the tethered C-R plasmid, followed by structural or functional examination of neurons.I would like to state that these experiments are not essential in my opinion, but any improvements in one of these directions would greatly improve and extend the impact and findings of this paper.

We thank the reviewer for the suggestion and the understanding. The suggested in vivo experiments are fascinating. However, in vivo imaging of dendritic spine morphology is already in itself challenging. The difficulty greatly increases when trying to detect partial, likely transient translocation of a signaling protein. It is also very difficult to knock down endogenous PKA while simultaneously expressing the R-C construct in a large number of cells to achieve detectable circuit or behavioral effect (and hope that compensation does not happen over weeks). We hope the reviewer agrees that these experiments would be their own project and go beyond the time and scope of the current study.

**Reviewer #3 (Recommendations For The Authors):**
Please elaborate on the methods used to visualize PKA-RIIα and PKA-RIβ subunits.

As suggested, we have now included additional details for visualizing PKA-Rs in the text. Specifically, we write (pg. 5): “…, as visualized using expressed PKA-R-mEGFP in separate experiments (Figs. 1A-1C).”.